# Protective Effects of Resveratrol on Adolescent Social Isolation-Induced Anxiety-Like Behaviors via Modulating Nucleus Accumbens Spine Plasticity and Mitochondrial Function in Female Rats

**DOI:** 10.3390/nu14214542

**Published:** 2022-10-28

**Authors:** Jinlan Zhao, Lihong Ye, Zuyi Liu, Yongfei Cui, Di Deng, Shasha Bai, Lei Yang, Yafei Shi, Zhongqiu Liu, Rong Zhang

**Affiliations:** 1Guangdong Provincial Key Laboratory of Translational Cancer Research of Chinese Medicines, Joint International Research Laboratory of Translational Cancer Research of Chinese Medicines, International Institute for Translational Chinese Medicine, School of Pharmaceutical Sciences, Guangzhou University of Chinese Medicine, Guangzhou 510006, China; 2School of Fundamental Medical Science, Guangzhou University of Chinese Medicine, Guangzhou 510006, China

**Keywords:** resveratrol, adolescent social isolation, nucleus accumbens, spine plasticity

## Abstract

Social isolation (SI) is a major risk factor for mood disorders in adolescents. The nucleus accumbens (NAc) is an important reward center implicated in psychiatric disorders. Resveratrol (RSV) is one of the most effective natural polyphenols with anti-anxiety and depression effects. However, little is known about the therapeutic effects and mechanisms of RSV on behavioral abnormality of adolescent social stress. Therefore, this study aimed to investigate the underlying mechanism of RSV on the amelioration of SI-induced behavioral abnormality. We found that SI induced anxiety-like behavior and social dysfunction in isolated female rats. Moreover, SI reduced mitochondrial number and ATP levels and increased thin spine density in the NAc. RNA sequencing results showed that SI changed the transcription pattern in the NAc, including 519 upregulated genes and 610 downregulated genes, especially those related to mitochondrial function. Importantly, RSV ameliorated behavioral and spine abnormalities induced by SI and increased NAc ATP levels and mitochondria number. Furthermore, RSV increased the activity of cytochrome C oxidase (COX) and upregulated mRNA levels of Cox5a, Cox6a1 and Cox7c. These results demonstrate that the modulation of spine plasticity and mitochondrial function in the NAc by RSV has a therapeutic effect on mood disorders induced by social isolation.

## 1. Introduction

Early-life stress (e.g., child maltreatment, neglect and social isolation (SI)) is a major risk factor for depression, anxiety and substance abuse disorders [1]. In the special context of the COVID-19 pandemic, people around the world may need to keep a certain degree of social distance and reduce social interaction. Research shows that major depression and anxiety disorders increased by 28% and 26%, respectively, worldwide in 2020, with women and young people being the most affected [2,3]. Adolescence is characterized by increased sensitivity to social stimuli and social interaction with peers [4]. Research suggests that adolescents, especially girls, are more likely to suffer from depression and anxiety during and after isolation [3,5]. Recent studies provide strong evidence that adolescent social isolation in male rats increased the susceptibility to anxiety and addiction [4,6,7,8,9,10]. However, only a few studies have evaluated the effects of adolescent stress on anxiety and depression-like behaviors in female rodents [4,9,11], and little is known about the neurobiology mechanism underlying the behavioral abnormality in response to adolescent social-isolation stress in female rats.

As a key reward center, the nucleus accumbens (NAc) integrates cognitive and emotional information by receiving glutamatergic input from the prefrontal cortex, amygdala and hippocampus and dopaminergic input from the ventral tegmental area [12]. It is reported that early life stress has a significant effect on the NAc [1,13]. Medium spiny neurons (MSNs), the major cell type in the NAc containing a great deal of dendritic spines, have been linked to neuropsychiatric disease [12]. Evidence in rodent models showed that early life stress changed dendritic morphology in the NAc and led to NAc transcriptome changes which increase susceptibility to develop depression-related behavioral abnormalities [14]. Additionally, increasing evidence indicates that mitochondrial abnormalities in the NAc play a critical role in the pathophysiology of anxiety or depression-like behavior [15,16,17]. For example, highly anxious animals showed alterations in mitochondria function and spine density in the NAc, and overexpression of mitofusin 2 in the NAc reversed behavioral anomalies and associated mitochondrial and neuronal phenotypes [15]. It is worth noting that adolescence is a critical developmental stage of the NAc [4,9]. It has been reported that compared with healthy adolescents, adolescents with anxiety or depression showed smaller NAc volumes and reduced NAc response to reward stimuli [18]. Importantly, recent studies demonstrated that adolescent social isolation disrupted NAc glutamatergic transmission and transcriptional responses to cocaine [9,19]. In addition, Novoa J reported that adolescent social isolation increased anxiety-like behaviors and NAc dopamine release in male rats [7]. Taken together, these data suggest that targeting the structural and functional alterations in the NAc could provide a means of ameliorating mental disorders caused by adolescent social isolation.

Resveratrol (RSV), an effective natural polyphenol, is widely found in grapes, peanuts, blueberry, pomegranate, polygonum cuspidatum, and in other medicinal plants [20]. It has been reported that resveratrol has anti-oxidation, anti-inflammation, anti-platelet aggregation, anti-aging and anti-tumor properties and plays a critical role in modulating atherosclerosis, cardiovascular, cerebrovascular and age-related diseases, such as Parkinson’s disease and Alzheimer’s disease [20,21]. In recent years, much attention has been paid to the role of resveratrol in the nervous system. The protective effects of resveratrol on neurodegenerative diseases are related to the different processes and molecular events, such as the reduction in oxidative stress and the regulation of inflammation, mitochondrial function, apoptosis, and epigenetic modifications [20]. In addition to neurodegenerative diseases, increasing evidence demonstrates that resveratrol has antidepressant- and anxiolytic-like effects [22,23,24,25]. For instance, it was reported that RSV could alleviate depressive-like behaviors induced by estrogen deficiency and chronic unpredictable mild stress [26,27]. Another study reported that resveratrol could ameliorate social defeat stress-induced depressive-like behavior and locus coeruleus neuroinflammation [28]. Recent studies also demonstrated that mitochondrial dysfunction, neuroinflammation, oxidative stress and monoamine neurotransmitter changes play an important role in the pathology of social stress-induced depression [27,29]. In addition, antidepressants, antioxidants, antipsychotics and herbal medications have proven helpful in ameliorating the side effects of social isolation [30]; however, to our knowledge, little is known about the therapeutic effect and potential mechanism of resveratrol on adolescent depression and anxiety induced by social isolation.

Here, we evaluated the effects of resveratrol on adolescent social-isolation-induced behavioral deficits and investigated possible neurobiological mechanisms underlying resveratrol effects in the NAc. We demonstrated that adolescent social-isolation-induced anxiety-like behavior and social dysfunction in female rats. Furthermore, resveratrol improved spine and behavioral deficits induced by adolescent social isolation, accompanied by a reversal of stress-induced reduction in ATP levels, mitochondria number and mRNA levels of Cox5a, Cox6a1 and Cox7c. Taken together, our findings suggest that resveratrol is a potential drug for the treatment of anxiety-like behavior and social deficits induced by social isolation.

## 2. Materials and Methods

### 2.1. Experimental Materials

Resveratrol (LOT#501-36-0) was purchased from Sigma Chemical Company (St. Louis, MO, USA). Dimethyl sulfoxide (DMSO, LOT#67-68-5) was obtained from MP Biomedicals Biological Medicine Company (Irvine, CA, USA); 0.9% saline (LOT#H21080105) was purchased from Industry Group Co., Ltd. (Sichuan, China); 4% paraformaldehyde (LOT#G1101) and electron microscope fixative (LOT#G1102) were obtained from Servicebio Biotechnology Co., Ltd. (Wuhan, China). Sucrose (LOT#57-50-1) was purchased from Macklin Biotechnology Technology Co., Ltd. (Shanghai, China). Recombinant adeno-associated virus serotype 2/9 expressing EGFP (AAV-eGFP) driven by the hsyn promoter (LOT#PT5701/PT0933) was obtained from Brain VTA Technology Co., Ltd. (Wuhan, China). TriZol reagent (LOT#9109) was purchased from Invitrogen Biotechnology Co., Ltd. (Waltham, MA, USA). PrimeScript RT reagent Kit (LOT#AG11706) and SYBR Green reagent (LOT#AG11701) were obtained from Accurate Biotechnology Co., Ltd. (Hunan, China). All primers were purchased from Sangon Biotech Co., Ltd. (Shanghai, China). ATP Assay Kit (LOT#A095-1-1) was obtained from Nanjing Jiancheng Bioengineering Institute (Nanjing, China). Cytochrome-c oxidase activity assay kit (LOT#GMS10014.3.2) was purchased from Genmed Scientifics Company (Shanghai, China). Phosphate buffered saline (PBS, LOT#8120388) was obtained from Gibco Company (NYC, New York, NY, USA).

### 2.2. Animal Experiment

Twenty-one-day-old female Sprague–Dawley (SD) rats, weighted 46.78 ± 5.35 g, were obtained from the Animal Experimental Center of Guangzhou University of Chinese Medicine. On postnatal day (PD) 21, female rats were group-housed (4–5 rats per cage) or single-housed at a constant temperature (22–24 °C) and were allowed access to food and water. In addition, all animals were exposed to 12 h light/dark cycles. Behavioral tests were conducted after animals were housed under isolation or social conditions for 2 or 5 weeks at mid-adolescence or late adolescence (see Figure 1A for a brief study timeline). In general, the majority of data indicates that P21–P34 correspond to early adolescence, P34–P46 to mid-adolescence, and P46–P59 to early adulthood or late adolescence [11,31,32]. The rats were still raised in groups or in isolation during the behavioral tests.

### 2.3. Drug Treatment

Resveratrol was dissolved in DMSO and diluted in saline at a concentration of 20 mg/mL. RSV was administered via the intraperitoneal at a dose of 20 mg/kg based on previous studies [26]. DMSO (as vehicle control) or 20 mg/kg RSV was administered from PD35 to PD61. All drugs were freshly prepared.

### 2.4. Sucrose Preference Test (SPT) 

Anhedonia is measured by a sucrose preference test. The sucrose preference test was performed according to our previous study with some adjustments [33]. On the first day, two bottles of 1% sucrose solution were given to the rats for acclimatization; on the second day, the rats were exposed to a bottle of 1% sucrose solution and a bottle of water for 24 h, and the positions of the two bottles were changed after 12 h; on the third day, the rats were fasted for 24 h; on the fourth day, a bottle of 1% sucrose solution and drinking water were given to the rats for 2 h, and the positions of the two bottles were changed after 1 h of testing. The difference between the weight before and after the drinks was used to measure sucrose solution intake. Sugar preference is calculated as (sucrose consumption divided by water consumption plus sucrose consumption) × 100.

### 2.5. Open Field Test (OFT)

Rats were gently placed in the middle of a square box (100 cm × 100 cm × 50 cm), and the activity of rats in the open field arena was recorded by an automatic tracking and analysis system for 10 min. The locomotor activity of rats was measured by the total distance, and anxiety-like behavior was measured by the percentages of time spent in the center of the open field arena [12].

### 2.6. Elevated plus Maze (EPM)

The elevated plus maze assay was performed as previously described [11,34]. Briefly, the maze consisted of two closed arms (50 × 10 × 40 cms) and two open arms (50 × 10 cms) which were elevated 50 cm above the floor. The intersection between the open arm and the closed arm was an open square platform (10 × 10 cms), and there was a real-time surveillance camera above the square platform for video capture. The rats were gently placed into the middle of the maze and allowed to explore the maze for 5 min freely. Subsequently, anxiety-like behavior was measured by an anxiety index calculated as: 1-[(Open arm entry times/Total arm entry times) + (Open arm duration/Total arm duration)]/2 [11].

### 2.7. Forced Swim Test (FST)

The FST was performed according to our previous study [33]. The immobility times of the rats were recorded for the entire period for 5 min. The immobility time is defined as the time during which the rat floats or keeps only a slight limb movement in order to maintain balance.

### 2.8. Social Interaction Test (SIT)

Social interaction behavior was evaluated by the three-chamber test according to a previous study [35]. The apparatus consisted of three compartments, with two side chambers connected to a central chamber. Each lateral chamber had a small cylindrical cage. Female rats were allowed to explore three compartments freely for 10 min before being confined to the central chamber. An unfamiliar stimulus rat who had never come into contact with the experimental rats was randomly put into the cage in the left or right chamber while the cage was left empty in another chamber. Subsequently, the glass resin plates between the chambers were taken out, so that the experimental rat could explore the three compartments for 10 min freely. Social interaction behavior was measured by discrimination index: (time spent sniffing the stimulus rat-time spent exploring the empty chamber)/(total time spent exploring both the stimulus rat and the empty chamber) × 100. 

### 2.9. Viral Constructs and Microinjection

Recombinant adeno-associated virus serotype 2/9 expressing eGFP (AAV-eGFP) driven by the hsyn promoter were purchased from Brain VTA (Wuhan, China). The anaesthetized rat was placed in the stereotaxic apparatus, and the viruses (0.4 μL/side) were bilaterally infused into the NAc over 5 min at the coordinates AP, +0.96 mm; ML, ±1.50 mm; and DV, −7.40 mm [12,36]. To allow the virus to diffuse, the needle remained in place for another two minutes.

### 2.10. Dendritic Spine Analysis

To observe the effects of social isolation and resveratrol on spine remodelling in the NAc, AAV-eGFP were bilaterally infused into the NAc at PD 21. Twenty-four hours after behavioral tests, the rats were sacrificed by transcardial perfusion with phosphate-buffered saline and 4% paraformaldehyde. The NAc sections were collected from the rat and immunostained with anti-GFP antibody (1:500). Then, an Alexa Fluor 488-conjugated anti-rabbit antibody (1:200) was applied. A laser confocal microscope (LEICA, DMIRE2, and Germany) was used to capture all images of the spines. We classified dendritic spines in three categories: thin spines (spine head < 0.45 μm), mushroom spines (spine head > 0.45 μm) and stubby spines defined as protrusions without neck. Spine analysis was performed according to previous study [12,15]. 

### 2.11. RNA-Seq Analysis

Twenty-four hours after behavioral tests, the NAc sections were collected from rats according to our previous study [12]. Female NAc samples were pooled into groups of 2 animals/sample (*n* = 6 per group). The NAc sections were sent to the Nanjing Tongyuan Medical Laboratory to prepare libraries for sequencing using Illumina NovaSeq 6000. DESeq2 was used to detect the significant differentially expressed genes (DEGs) with Benjamini–Hochberg corrected *p* < 0.05. Gene Ontology (GO) and the Kyoto Encyclopedia of Gene and Genomes (KEGG) databases were used for enrichment analysis of the downregulated and upregulated gene sets.

### 2.12. Quantitative Real-Time PCR (qPCR)

According to the manufacturer’s instructions, total RNA was isolated with TriZol reagent. The PrimeScript RT reagent kit was used to synthesize the cDNA, and qPCR amplification was performed using SYBR Green reagent by using the QuantStudio 5 (Applied Biosystems, Waltham, MA, USA). Analysis was performed using the ΔΔC(t) method. All the primers were provided by Sangon Biotech (Shanghai, China). Cox5a, 5′–CCATTCGCTGCTATTCTCATGG–3′ (Forward, F); 5′–AAGTATGTCACCCAGCGAGC–3′ (Reverse, R); Cox6a1, 5′–GAGGAGGGTTCAGCTCGTATT–3′ (F); 5′–TTCATAGCCAGTCGGAAGCG–3′ (R); Cox7c, 5′–ACAAGTGGCGGTTACTGCTTAT–3′ (F); 5′–ACTATAAAGAAAGGAGCAGCAAATC–3′ (R); Gapdh, 5′–ACCACAGTCCATGCCATCAC–3′ (F); 5′–TCCACCACCCTGTTGCTGTA–3′ (R). 

### 2.13. Transmission Electron Microscopy

Twenty-four hours after behavioral tests, the NAc samples were collected and immediately placed in an electron microscope fixative and fixed at 4° for 2–4 h. Then, the NAc sample was sent to the Servicebio Technology Co., Ltd., Gent, Belgia. After the sample was prepared, the ultrastructure of the NAc was observed and photographed using a transmission electron microscope (HT7800, Hitachi, Japan).

### 2.14. Cytochrome-c Oxidase (COX) Activity and Adenosine Triphosphate Content

A Cytochrome-c oxidase activity assay kit was used to measure the activity of cytochrome C oxidase in the NAc and expressed as U/mg protein [37]. The NAc adenosine triphosphate (ATP) content was assessed using an ATP Assay Kit according to the manufacturer’s instructions [38]. 

### 2.15. Statistical Analysis

SPSS 22.0 was used for statistical analysis. We analyzed the results of two groups using Student’s *t*-tests and the results of more than two groups using one-way ANOVA followed by Bonferroni’s post hoc test. Data are presented as the mean ± SEM. Significance was set at *p* < 0.05.

## 3. Results

### 3.1. Social Isolation Induced Anxiety-Like Behavior and Social Dysfunction but Not Depression-Like Behavior at Late Adolescence in Female Rats

Numerous studies have illustrated that timing of social isolation exposure may affect behavioral outcomes [4]. In this experiment, female rats were isolated after weaning, and then the behavioral changes including sucrose preference, open field test, elevated plus maze, forced swim and social interaction tests in middle adolescence and late adolescence were detected (Figure 1A). 

Depressive-like behaviors were evaluated by the SPT, FST and SIT; the anxiety-like behaviors were monitored by OFT and EPM. As shown in Figure 1, the percentage of sucrose consumed (*n* = 15, t28 = 0.458, *p* = 0.650, Figure 1B) showed no difference between isolated rats and group-housed rats. In the open field test, there was no difference in the total distance (*n* = 15, t28 = −1.145, *p* = 0.262, Figure 1C) and the time spent in the central area (*n* = 15, t28 = −0.899, *p* = 0.376, Figure 1D) of the isolated rats compared with the control group. Moreover, immobility time (*n* = 15, t28 = −0.011, *p* = 0.991, Figure 1E) and the percentage of social interaction (*n* = 15, t28 = −0.276, *p* = 0.784t, Figure 1F) showed no difference between isolated rats and group-housed rats. In the EPM, the percentage of anxiety index of isolated mice did not change compared with the control group (*n* = 15, t28 = −0.592, *p* = 0.599, Figure 1G). These results indicated that social isolation (P21–P35) had no effect on depression- and anxiety-like behaviors compared with the control group at middle adolescence. Interestingly, we found that social isolation in female rats induced anxiety-like behavior rather than depression-like behavior at late adolescence. There were no significant changes in the percentage of sucrose consumed (*n* = 15, t28 = 0.364, *p* = 0.718, Figure 1H) and immobility time (*n* = 15, t28 = 1.029, *p* = 0.312, Figure 1K). The isolated rats showed less time in the middle of the open field (*n* = 15, t28 = 2.282, *p* = 0.030, Figure 1J), but the total distance did not change (*n* = 15, t28 = 1.049, *p* = 0.303, Figure 1I). In addition, the isolated rats exhibited decreased percentage of social interaction (*n* = 15, t28 = 7.821, *p* < 0.001, Figure 1L) and increased anxiety behavior as assessed by the percentage of anxiety index in the EPM (*n* = 15, t28 = −6.054, *p* < 0.001, Figure 1M). Taken together, these data suggested that the timing of social isolation exposure was important to behavioral outcomes and that prolonged social isolation could lead to anxiety-like behavior rather than depression-like behavior in female rats at late adolescence. 

### 3.2. The Effect of Adolescent Social Isolation on Spine Remodelling and Mitochondrial Function in the NAc

Emerging evidence points to a central role of spine structural plasticity in psychiatric disorders [12,39]. Thus, we bilaterally infused AAV-eGFP control viruses into the NAc to examine the effect of social isolation on spine plasticity in the NAc. After the virus was fully expressed, behavioral tests were conducted. Twenty-four hours after the test, NAc brain tissues were collected for dendritic spine analysis (Figure 2A). The results showed that social isolation increased the total spine density (8–12 dendrite sections per animal with 3–4 animals per group, t83 = −5.822, *p* < 0.001, Figure 2C) compared to group-housed rats, and social isolation also increased thin spine density (t83 = −8.180, *p* < 0.001, Figure 2D). As shown in Figure 2E,F, social isolation had no effect on the density of mushroom (t83 = 0.504, *p* = 0.616) and stubby spines (t83 = 1.896, *p* = 0.061) in the NAc. 

Given that mitochondria have been implicated in anxiety behavior and synapse plasticity [15,40], we further observed the structural changes of mitochondria using electron microscopy. As shown in Figure 2G, in the control group, the morphology of mitochondria was normal with irregular cristae. In contrast, morphological abnormalities with incomplete outer membrane and altered arrangement of the cristae were observed in isolated rats. Additionally, some mitochondria were swollen, showing pale matrix with very few cristae in isolated rats. Moreover, the number of mitochondria per unit area (μm^2^) was significantly reduced in the isolated rats (*n* = 4, t6 = 3.103, *p* = 0.021, Figure 2H), and the isolated rats displayed lower ATP content than control rats (*n* = 5, t8 = 4.323, *p* = 0.003, Figure 2I). Together, these results indicate that the isolated rats showed increased thin spine density, as well as damaged mitochondrial structure and function in the NAc.

### 3.3. The Effect of Adolescent Social Isolation on the Transcriptional Patterns in the NAc

To better understand how the NAc is involved in social-isolation-induced behavioral abnormality in female rats, we investigated transcriptomic changes in the NAc of female rats. An adjusted *p*-value < 0.05 was selected as the threshold value for the differentially expressed genes (DEGs) screening. As shown in Figure 3A, RNA-seq revealed 519 upregulated genes and 610 downregulated genes in the NAc of isolated females, compared to the NAc of group-housed female rats. GO terms and KEGG pathway enrichment analysis were conducted to gain further insight into the biological functions of DEGs identified. 

According to the KEGG pathway enrichment analysis, upregulated DEGs mainly participated in neuroactive ligand–receptor interaction, GABAergic synapse, morphine addiction, nicotine addiction, calcium signaling pathway, glutamatergic synapse, retrograde endocannabinoid signalling, gap junction, axon guidance and cocaine addiction. The KEGG analysis of downregulated DEGs were significantly enriched in ribosome, oxidative phosphorylation, Parkinson’s disease, Huntington’s disease, Alzheimer’s disease, thermogenesis, non-alcoholic fatty liver disease (NAFLD) and cardiac muscle contraction (Figure 3B,C). GO analysis results showed that the upregulated DEGs were significantly enriched in learning, locomotory behavior, associative learning, modulation of chemical synaptic transmission learning or memory, cognition, response to ammonium ion, regulation of neurotransmitter levels, negative regulation of synaptic transmission and neurotransmitter transport (Figure 3D). The GO result of downregulated DEGs were significantly enriched in the electron transport chain, purine ribonucleoside triphosphate metabolic process, mitochondrial ATP synthesis coupled electron transport ATP metabolic process, ribonucleoside triphosphate metabolic process and so on (Figure 3E), which suggested that the mitochondrial function was significantly impaired. In order to identify the key proteins and important protein modules of mitochondrial damage caused by social stress, we carried out protein–protein interaction (PPI) analysis. The PPI network of the top 100 downregulated genes in the isolated rats was obtained with the help of the STRING database (http://www.string-db.org/ (accessed on 6 June 2022)). We found that Cox5a, Cox6a1 and Cox7c were significantly enriched in mitochondrial electron transport (red), cytochrome-c oxidase (COX) activity (blue) and respiratory chain complex iv (green) (Figure 3F), indicating that these genes could be associated with social-isolation-induced-mitochondrial dysfunction in the NAc.

### 3.4. The Protective Effects of Resveratrol on Adolescent Social-Isolation-Induced Behavioral Abnormalities in Female Rats

Resveratrol, the most effective natural polyphenol, has been found to improve anxiety-like behaviors [22,24]. Thus, we further evaluated whether resveratrol could alleviate behavior deficits induced by adolescent social isolation. Our results demonstrated that administration of resveratrol (20 mg/kg) produced significant alterations in anxiety-like behaviors when compared with vehicle-treated rats. As shown in Figure 4B, the total distance travelled was not altered by resveratrol in female rats (*n* = 12, F (2, 33) = 1.487, *p* = 0.241). In the OFT, the resveratrol-treated rats showed more time in the middle of the open field (*n* = 12, F (2, 33) = 16.005, *p* < 0.001, Bonferroni’s post hoc test: *p* < 0.001, SI+vehicle vs. Ctrl +vehicle group; *p* = 0.006, SI+RSV vs. SI+vehicle group, Figure 4C). In the EPM, resveratrol treatment significantly reduced the anxiety index when compared to vehicle-treated rats (*n* = 12, F (2, 33) = 6.874, *p* = 0.003, Bonferroni’s post hoc test: *p* = 0.016, SI+vehicle vs. Ctrl +vehicle group; *p* = 0.005, SI+RSV vs. SI+vehicle group, Figure 4D). In addition, one-way ANOVA analysis revealed that resveratrol could improve social deficits induced by adolescent social isolation in female rats (*n* = 12, F (2, 33) = 7.846, *p* = 0.002, Bonferroni’s post hoc test: *p* = 0.001, SI+vehicle vs. Ctrl+vehicle group; *p* = 0.047, SI+RSV vs. SI+vehicle group, Figure 4E).

### 3.5. The Protective Effects of Resveratrol on Adolescent Social Isolation-Induced Mitochondrial and Spine Abnormalities in Female Rats

Increasing evidence suggests that mitochondrial structural and functional changes as well as spine density in the NAc is critical for development of anxiety-like behavior and social dysfunction [15]. Thus, we further evaluated whether resveratrol could improve spine deficits and mitochondrial abnormalities induced by social isolation. We first measured the effect of resveratrol on mRNA levels of Cox5a, Cox6a1 and Cox7c in the NAc. We found that the Cox5a mRNA expression in the NAc varied significantly among the groups (*n* = 4, F (2, 9) = 228.208, *p* < 0.001, Figure 5A). Post hoc analyses demonstrated that the Cox5a was decreased in the isolated rats treated with vehicle (*p* < 0.001), and the RSV can significantly increase the mRNA levels of Cox5a (*p* < 0.001). Moreover, we found that resveratrol led to a significant increase in Cox6a1 (*n* = 4, F (2, 9) = 76.021, *p* < 0.001, Bonferroni’s post hoc test: *p* = 0.009, SI+vehicle vs. Ctrl+vehicle group; *p* < 0.001, SI+RSV vs. SI+vehicle group, Figure 5B) and Cox7c expression (*n* = 4, F (2, 9) = 286.287, *p* < 0.001, Bonferroni’s post hoc test: *p* < 0.001, SI+vehicle vs. Ctrl+vehicle group; *p* < 0.001 and SI+RSV vs. SI+vehicle group, Figure 5C) in the NAc. 

There was a significant correlation between the mRNA expression levels of COX-related genes and the activity of cytochrome C oxidase, which is not only the terminal enzyme of mitochondrial respiratory chain but also an important part of electron transport chain [41]. The change of cytochrome C oxidase activity can lead to the disorder of electron transport and the decrease of ATP production. As shown in Figure 5D–F, our results also demonstrated that resveratrol could significantly increase the activity of cytochrome C oxidase (*n* = 4, F (2, 9) = 16.222, *p* = 0.001, Bonferroni’s post hoc test: *p* = 0.003, SI+vehicle vs. Ctrl+vehicle group; *p* = 0.002, SI+RSV vs. SI+vehicle group, Figure 5D) and ATP content (*n* = 5, F (2, 12) = 9.468, *p* = 0.003, Bonferroni’s post hoc test: *p* = 0.018, SI+vehicle vs. Ctrl+vehicle group; *p* = 0.004, SI+RSV vs. SI+vehicle group, Figure 5E) in the NAc. In view of the close relationship between the morphology and function of mitochondria, we further performed electron microscopic analysis in the NAc of resveratrol-treated rats. As shown in Figure 5F, the morphology of mitochondria was normal in resveratrol-treated rats compared to vehicle-treated rats with incomplete outer membrane and altered arrangement of the cristae. Moreover, resveratrol significantly increases the number of mitochondria in the NAc (*n* = 4, F (2, 9) = 6.451, *p* = 0.018, Bonferroni’s post hoc test: *p* = 0.048, SI+vehicle vs. Ctrl+vehicle group; *p* = 0.030, SI+RSV vs. SI+vehicle group, Figure 5G). Finally, in addition to confirming the protective role of resveratrol in adolescent social-isolation-induced mitochondrial abnormalities, we also evaluated the role of resveratrol in NAc spine plasticity. We found that resveratrol reversed the increases in total spine density (F (2, 102) = 32.254, *p* < 0.001, Bonferroni’s post hoc test: *p* < 0.001, SI+vehicle vs. Ctrl+vehicle group; *p* < 0.001, SI+RSV vs. SI+vehicle group, Figure 5I) and thin spine density (F (2, 102) = 22.582, *p* < 0.001, Bonferroni’s post hoc test: *p* < 0.001, SI+vehicle vs. Ctrl+vehicle group; *p* < 0.001, SI+RSV vs. SI+vehicle group, Figure 5J) induced by adolescent social isolation. Additionally, our results demonstrated that resveratrol had no effect on the mushroom spine density (F (2, 102) = 1.522, *p* = 0.223, Figure 5K) and stubby spine density (F (2, 102) = 0.084, *p* = 0.920, Figure 5L).

## 4. Discussion

Adolescence is considered to be a developmental period of vulnerability to stress. During the COVID-19 pandemic, depression and anxiety symptoms increased among adolescents, particularly among girls [3]. However, most rodent research was conducted in adult male animals [4,6,7,8,9,10], despite the increased anxiety and depression prevalence in women and young people [3,5,11]. Our results indicate that anxiety-like behaviors and social deficits are particularly sensitive to adolescent social isolation in female rats and reveals differences in spine plasticity and mitochondrial function in the NAc as a potential mechanism underlying the susceptibility to anxiety-like and social interaction behavior. Additionally, we found that resveratrol improved spine and behavioral deficits induced by adolescent social isolation, indicating that resveratrol is a potential drug for the treatment of mood disorders induced by adolescent social isolation.

Numerous studies have examined the effects of adolescent social isolation on adult behavior [7,9,32], while few studies have evaluated the effects of adolescent social isolation on anxiety and depression-like behavior in adolescence. D’Souza D [11] used female adolescent Wistar Kyoto rats and age-matched Wistars to evaluate the effects of age and species on anxiety and depression-like behavior and found that female Wistar Kyoto rats showed anxiety-like behavior during early adolescence and exhibited depression-like behavior during early and mid-adolescence, suggesting that age and strain created significant effects on behavioral results. Thus, in this experiment, we evaluated the anxiety and depressive-like behavior during mid- and late adolescence. We found that social isolation from PD21 to PD34 had no effects on anxiety and depression-like behavior as well as social behavior in female Sprague–Dawley rats when tested at PD35. Furthermore, social isolation from PD21 to PD55 induced anxiety-like behavior and social dysfunction but not depression-like behavior, suggesting that the timing of social isolation exposure is important for the behavioral results and mid- and late adolescence isolation represents the critical period for the effects of social isolation on anxiety-like behaviors and social dysfunction in female Sprague–Dawley rats. Our study is in accordance with a previous study showing that social isolation increased anxiety-like behavior in male Long–Evans rats during adulthood [34]. However, it is worth noting that our results contrast with a previous study in that 6 weeks of social isolation induced both depression-like and anxiety-like behaviors in male C57BL/6 mice [42]. In our study, the SI female rats only exhibited anxiety-like behaviors. This discrepancy may be attributed to differences between gender and species used in our experiments, since a large number of studies have shown that gender and species have a great influence on behavioral results [11,43]. Over time, evidence suggests that social isolation in adolescence has long-term influences on anxiety-like behavior and social interactions in female rodents [32,43]. For example, a recent study reported that after 5 weeks of post-weaning social isolation, female mice exhibited anxiety-like behaviors and social withdrawal [43]. Additionally, 3 weeks of post-weaning social isolation had no effect on anxiety-like behaviors in female rats [10]. Moreover, our results are echoed by previous studies showing that social isolation in adolescence from PD30 to PD50 had no effect on depression-like behavior in female rats [44,45]. Thus, it is important to note that because of differences in separation length, behavioral testing timing and experimental design, social-isolation stress may have inconsistent behavioral results [4]. Taken together, our study provides further evidence of the detrimental effects of adolescent social isolation in female rats and indicates that anxiety-like behaviors but not depression-like behaviors are particularly sensitive to adolescent social isolation in female rats.

Adolescence is a critical developmental stage of the glutamatergic and dopaminergic systems in the NAc [4]. The dendritic spine of the NAc is the target for the convergence of dopaminergic axons from the ventral tegmental area and glutamatergic axons from the prefrontal cortex, amygdala and hippocampus [12]. Therefore, it is important to evaluate the effect of social isolation in adolescence on the remodeling of NAc dendritic spines. In our study, we found that social isolation increased NAc total spine density, especially the density of thin spines, but had no effect on the density of mushroom and stubby spines. It is worth noting that, unlike the increase in total spine density found in our experiments, a previous study showed that highly anxious adult male Wistar rats exhibited lower total spine density compared to low- rats but displayed more thin spine density than low-anxiety rats [15]. This discrepancy may be due to the different anxiety models, anxiety levels, and different sex used in our experiments, since previous studies have shown sex differences in social-isolation-induced synaptic plasticity [11,46]. A previous study reported that resveratrol could attenuate depression-like behavior and the decrease of spine density in medial prefrontal cortex and hippocampus induced by chronic restraint stress [47]. In our study, we found that resveratrol administration can improve anxiety-like behavior and reduce the increase of dendritic spines in the NAc induced by social isolation, suggesting that resveratrol plays a region-specific regulatory effect on dendritic spines. Moreover, many studies have shown that the regulatory effect of stress on synaptic remodeling is different in brain regions [48]. It was reported that the number and morphology of spines are connected to synaptic plasticity and circuit remodeling in the NAc [12]. Thus, it is reasonable to speculate that the increased density of thin spines in female isolated rats may represent increasing synaptic connectivity from other brain regions. For instance, it was reported that the enhancement of ventral hippocampal (vHIP) glutamatergic afferents to the nucleus accumbens promoted susceptibility to chronic social stress, and the vHIP-NAc pathway was linked to anxiety-like and social interaction behavior [49,50]. Our transcriptome results also showed that upregulated DEGs mainly participated in neuroactive ligand–receptor interaction, GABAergic and glutamatergic synapse, morphine, nicotine and cocaine addiction. Additionally, it is important to mention that the increase in spine density and dopamine signaling found in the NAc has been linked to both rewarding and stressful stimuli [1,36]. In parallel to an increase of thin spine density in the NAc, the results of NAc transcriptome showed that the dopamine D1 and D2 receptor genes which have been associated with spine plasticity [36], significantly increased after adolescent social isolation. Karkhanis AN also demonstrated that social isolation increased nucleus accumbens dopamine and norepinephrine response to acute ethanol during adulthood, which is consistent with our transcriptome results showing that dopamine signaling pathway increased significantly after adolescent social isolation [34]. Additionally, our results are consistent with a previous report that adolescent social isolation increased the levels of D2 receptor expression in the NAc of adult rats [51]. Recent findings have also confirmed that social isolation in adolescent male rats increased anxiety-like behavior and the sensitivity of dopaminergic neurons in the NAc [3]. Moreover, our results are echoed by previous studies showing that social isolation in adolescence increases cocaine desire in both males and females [9,52]. Thus, we speculate that the increased thin spine density induced by adolescent social isolation might also be partially associated with the susceptibility to drug addiction in later life. Taken together, our results further support earlier studies that report that the “hyper-responsive dopaminergic system” in the NAc is closely related to early life adversity [34,51].

Increasing evidence implicates the involvement of mitochondrial dysfunction in the etiology of anxiety-like behaviors [15,40]. In the present study, we found that adolescent social isolation reduced NAc ATP levels and mitochondria number in female rats. Moreover, resveratrol could ameliorate anxiety-like behavior and social dysfunction induced by adolescent social isolation and increase NAc ATP levels as well as the number of mitochondria. Indeed, the anti-anxiety and antidepressant effects of resveratrol have been studied in a variety of animal models [28], and numerous studies demonstrated that resveratrol produces anxiolytic and antidepressant by regulating inflammation, oxidative stress, hypothalamic⁻pituitary⁻adrenal axis and neurogenesis [24,25,53]. For example, Sahin TD reported that resveratrol ameliorated streptozotocin-induced anxiety and depressive-like behaviors by inhibition of oxidative stress [24]. Another study reported that resveratrol could ameliorate social-defeat-stress-induced depressive-like behavior and locus coeruleus neuroinflammation [28]. A recent study also demonstrated that resveratrol alleviated anxiety-like behavior induced by lumbar spine instability surgery and relieved inflammation in the hippocampus [53]. Although a large number of studies have been carried out on the antidepressant mechanism of resveratrol [25], the therapeutic effect of resveratrol on adolescent depression and anxiety and the potential mechanism underlying resveratrol effects in the NAc remain largely unexplored. It is noteworthy that our study is the first to provide evidence that RSV produces an anti-anxiety effect by modulating nucleus accumbens spine plasticity and mitochondrial function in the context of adolescent social-isolation stress. Cytochrome C oxidase is not only the terminal enzyme of mitochondrial respiratory chain but is also an important part of the electron transport chain. The change in its function can lead to the disorder of electron transport and decrease ATP production [41,54]. Our transcriptome results also showed a significant decrease in mitochondrial-function-related genes, such as the COX-related genes. The levels of expression of the COX-related genes were correlated with the activity of cytochrome C oxidase and ATP production [41]. Consistent with these findings, we found that resveratrol increased the expression of COX genes (Cox5a, Cox6a1, and Cox7c) and the activity of the COX. Our results further illustrated the important role of COX-related genes the activity of the COX in mitochondrial function and provided preliminary evidence that resveratrol may exert an anxiolytic effect of social isolation by modulating COX gene expression and the activity of the COX in the NAc.

To the best of our knowledge, this study is the first to show that social isolation during adolescence could affect NAc spine plasticity and mitochondrial function in female rats and provide evidence that RSV produces an anti-anxiety effect via modulating nucleus accumbens spine plasticity and mitochondrial function in the context of adolescent social isolation stress. However, little is known about the precise mitochondria and spine remodeling mechanisms in social-isolation-stress-induced behavioral deficits and about how resveratrol improves spine and mitochondrial deficits in the NAc. It is important to note that one of the limitations of our study is that we only evaluated the effects of adolescent social isolation on the behavior of female rats, without comparing the sex differences in social-isolation-induced behavioral abnormalities. Further investigation is needed to explore behavioral and mechanism differences between males and females. Overall, our findings provide potential therapeutic targets for social-stress-related mental illness and suggest that resveratrol is a potential drug for the treatment of anxiety-like behavior and social deficits induced by adolescent social isolation.

## Figures and Tables

**Figure 1 nutrients-14-04542-f001:**
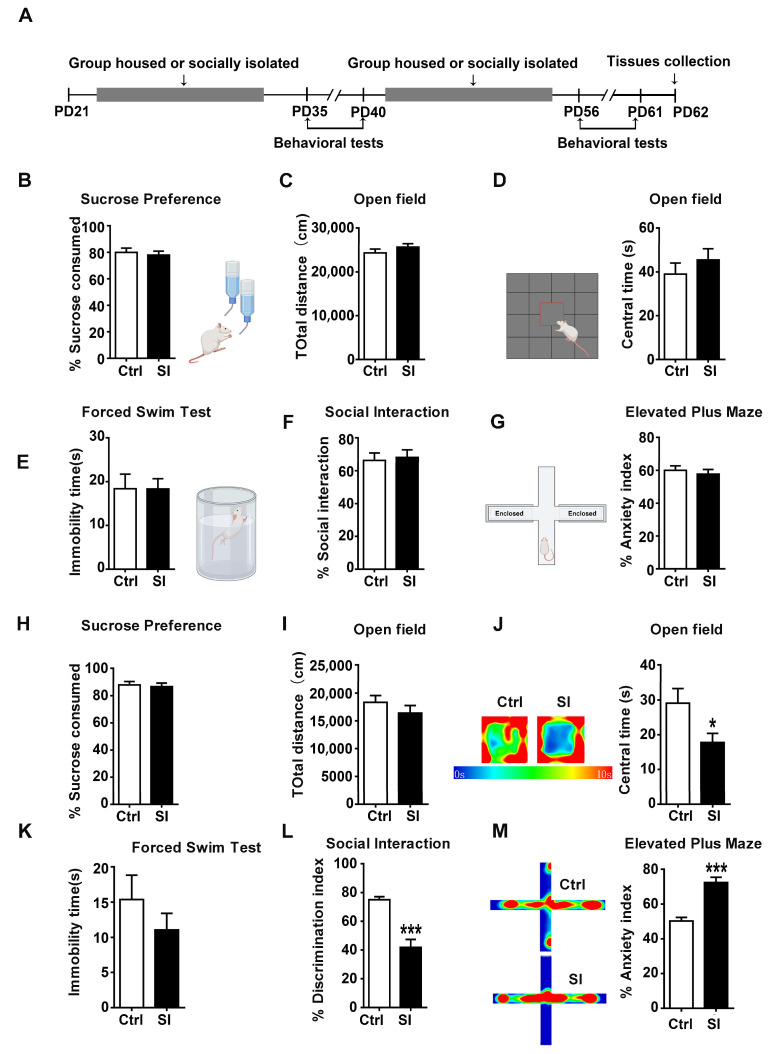
Social-isolation-induced anxiety-like behavior and social dysfunction but not depression-like behavior at late adolescence in female rats: (**A**) experimental paradigm for adolescent social isolation; (**B**) social isolation had no effect on the percentage of sucrose consumed at early adolescence; (**C**,**D**) in the open field test, adolescent social isolation had no effect on the total distance and time spent in the central area at early adolescence; (**E**–**G**) social isolation had no effect on the immobility time, the percentage of social interaction and the percentage of anxiety index at early adolescence; (**H**) social isolation had no effect on the percentage of sucrose consumed at late adolescence; (**I**) adolescent social isolation had no effect on the total distance at late adolescence; (**J**) the isolated rats showed less time in the middle of the open field at late adolescence; (**K**) social isolation had no effect on the immobility time at late adolescence; (**L**) the isolated rats exhibited decreased percentage of social interaction at late adolescence; (**M**) the isolated rats exhibited increased anxiety behavior as assessed by the percentage of anxiety index in the elevated plus maze at late adolescence; data were analysed using Student’s *t*-test and presented as mean ± SEM; *** *p* < 0.001 and * *p* < 0.05 compared to the isolated group; Ctrl indicates control; SI indicates social isolation.

**Figure 2 nutrients-14-04542-f002:**
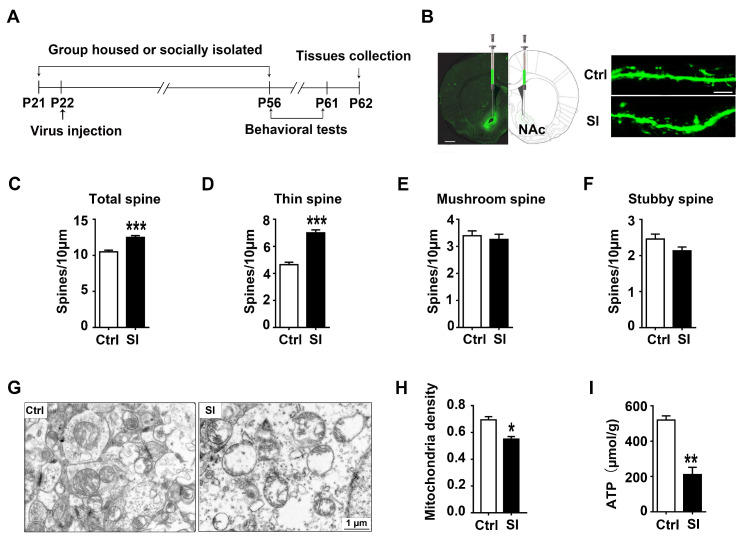
The effect of adolescent social isolation on spine remodeling and mitochondrial function in the NAc: (**A**) an experimental design for the injection of viruses and the collection of tissues; (**B**) representative confocal images of spines in female rats injected with viruses expressing eGFP; left image scale bar equals 1 mm, right image scale bar equals 7.5 µm; (**C**) adolescent social isolation significantly increased the total spine density; (**D**) adolescent social isolation significantly increased the density of thin spines; (**E**,**F**) adolescent social isolation had no effect on spine density of the mushroom and stubby; (**G**) images of control and isolated rats’ NAc obtained by electron microscopy; scale bars = 1 μm; (**H**) adolescent social isolation decreased the mitochondrial density of the NAc in isolated female rats; (**I**) adolescent social isolation decreased the ATP content of the NAc in isolated female rats. Student’s *t*-test was used for data analysis. Data were presented as mean ± SEM; *** *p* < 0.001; ** *p* < 0.01; and * *p* < 0.05 compared to the isolated group; Ctrl indicates control; SI indicates social isolation.

**Figure 3 nutrients-14-04542-f003:**
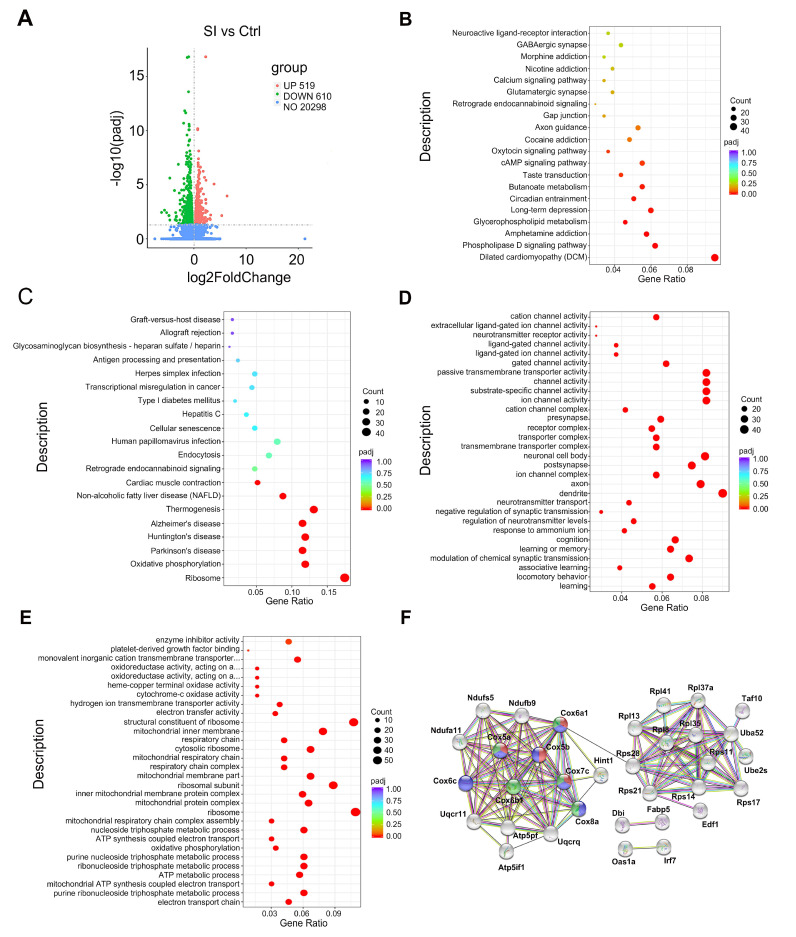
The effect of adolescent social isolation on the transcriptional patterns in the NAc: (**A**) volcano plots for differential expression genes. Genes that are upregulated are shown in red plots, while genes that are downregulated are illustrated in blue plots; (**B**,**C**) the KEGG analysis of the upregulated DEGs and downregulated DEGs in isolated female rats; (**D**,**E**) the GO analysis of the upregulated DEGs and downregulated DEGs; (**F**) analysis of the PPI network of the top 100 downregulated genes in the isolated rats. The nodes represent the downregulated genes in the PPI network, and the line colour indicates the type of interaction evidence (minimum required interaction score = 0.70); Ctrl indicates control; SI indicates social isolation.

**Figure 4 nutrients-14-04542-f004:**
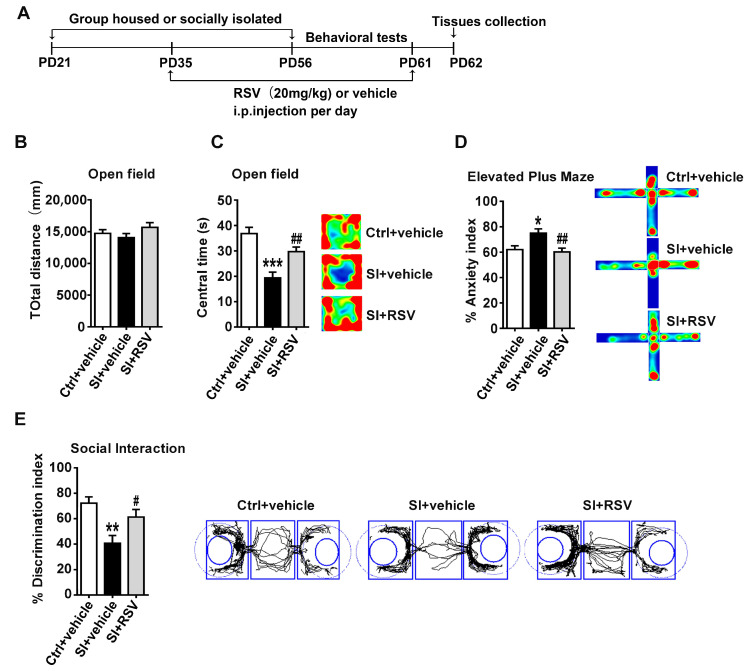
The protective effects of resveratrol on adolescent social-isolation-induced behavioral abnormalities in female rats: (**A**) experimental design for the resveratrol treatments procedure and tissues collection: (**B**,**C**) total distance and central time in the open test among all groups; (**D**) resveratrol treatment significantly reduced the anxiety index when compared to vehicle-treated isolated female rats; (**E**) resveratrol improved social deficits compared with the vehicle-treated isolated female rats; analyses were performed using one-way ANOVA followed by Bonferroni’s multiple comparison post hoc test; data were presented as mean ± SEM; *** *p* < 0.001; ** *p* < 0.01; and * *p* < 0.05 compared to the Ctrl +vehicle group; ## *p* < 0.01 and # *p* < 0.05 compared to the SI+vehicle groups; Ctrl indicates control; SI indicates social isolation.

**Figure 5 nutrients-14-04542-f005:**
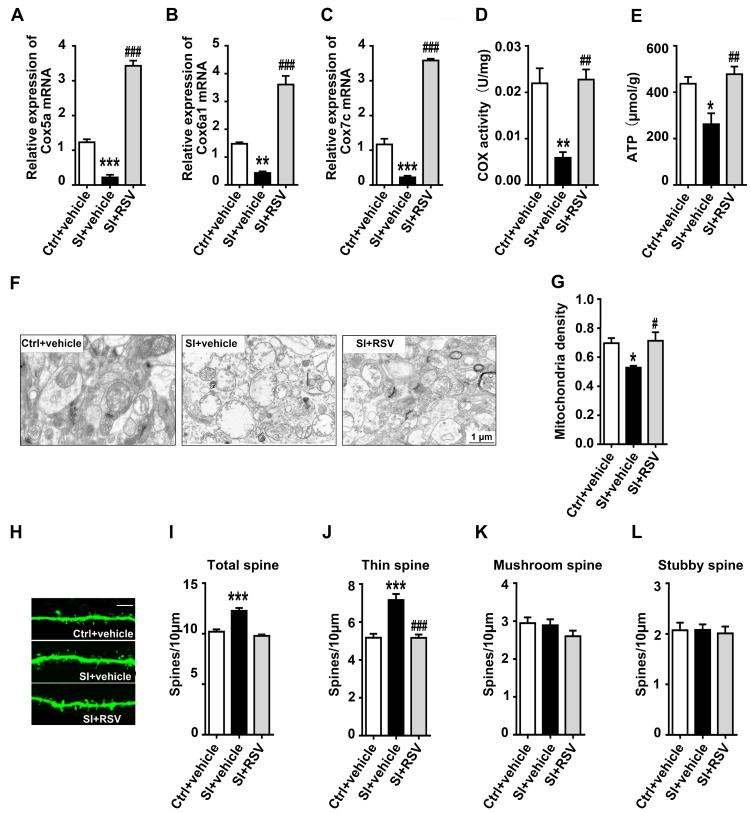
The protective effects of resveratrol on adolescent social-isolation-induced mitochondrial and spine abnormalities in female rats: (**A**–**C**) resveratrol treatments significantly increased the mRNA levels of Cox5a, Cox6a1 and Cox7c in the NAc compared with vehicle-treated isolated rats; (**D**) resveratrol treatments increased the activity of cytochrome C oxidase in the NAc; (**E**) resveratrol treatments increased the ATP content in the NAc; (**F**) representative electron micrographs among all groups; scale bars = 1 μm; (**G**) resveratrol increased the number of mitochondria in the NAc; (**H**) images of dendritic segments taken with confocal microscopy; scale bar = 5 µm; (**I**,**J**) resveratrol blocked the social-isolation-induced increase in the total and thin spine density compared with vehicle-treated SI rats; (**K**,**L**) no difference in mushroom and stubby spine density was observed among all groups; analyses were performed using one-way ANOVA followed by Bonferroni’s multiple comparison post hoc test; data were presented as mean ± SEM; *** *p* < 0.001; ** *p* < 0.01; and * *p* < 0.05 compared with the Ctrl +vehicle group; ### *p* < 0.001; ## *p* < 0.01; and # *p* < 0.05 compared with the SI+vehicle groups; Ctrl indicates control; SI indicates social isolation.

## Data Availability

The raw sequences data reported in this study are available at the NCBI under the BioProject PRJNA883947.

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
