# Peer review of "Protective Effects of Resveratrol on Adolescent Social Isolation-Induced Anxiety-Like Behaviors via Modulating Nucleus Accumbens Spine Plasticity and Mitochondrial Function in Female Rats"

_nutrients, 2022, doi:10.3390/nu14214542_

Round 1
Reviewer 1 Report
I have reviewed the paper entitled “Protective effects of resveratrol on adolescent social 2 isolation-induced anxiety-like behaviors via modulating 3 nucleus accumbens spine plasticity and mitochondrial function 4 in female rats”. The paper contains very good data however needs modification before publishing.
Comments
1. The key results should be included in the abstract section.
2. The introduction section is not decorated the authors are advised to include updated literature from SCOPUS.
3. Detail such as the sex, and weight o animals should be mentioned clearly in the experiment part.
4. The discussion par needs to be improved.
5. The grammatical issue should be fixed.
Reviewer 2 Report
The authors chose female rats for the experiments. Is there any reason for choosing female rats?
-there is no information on other resveratrol properties. More literature should be cited on resveratrol. the authors can use the ref. doi.org/10.3390/molecules27196225
line 98-99 "RSV was dissolved in DMSO. " What is the concentration of the prepared solution?
in the "2.3. Sucrose Preference Test" give more information about the test method.
The authors mentioned the methods by referring to ref 19, 20, and 21 in the method section. But the results were not compared with the refs. compare the RSV to the items in these refs.
On the basis of the manuscript, the authors must clarify the research space or gap that still needs to be addressed.
The authors mentioned the methods by referring to 19,20,21 refs in the method section, but the results were not compared with this ref. Compare the RSV to the items in these references.
Reviewer 3 Report
The authors have investigated the protective effects of resveratrol on adolescent social isolation-induced anxiety-like behaviors via modulating nucleus accumbens spine plasticity and mitochondrial function in female rats. The work is well organized and suitable for publication after some modifications suggested below:
Abstract: Quantitative results are missing in the abstract. Authors are suggested to include some quantitative results in order to enhance the readability of the manuscript.
The novelty of the work should be established.
Section 2: Materials and Methods, authors have directly started with methods. Materials section is missing. Kindly include materials section.
Figure 3: The letters are not clear.
The concussion should be concise and to the point indicating the application of the work.
Please improve the resolution and quality of all Figures.
Please compare your results with previous studies and mention clearly how your work is important in comparison to already been reported.
Avoid abbreviations before giving their explanation.
The authors should insert more recent publications within 2 years in References part and references should be revised due to duplicated and missing references.
Round 2
Reviewer 2 Report
it is acceptable.